# Aquatic Angiosperm Transplantation: A Tool for Environmental Management and Restoring in Transitional Water Systems

Adriano Sfriso [1,*], Alessandro Buosi [1], Yari Tomio [1], Abdul-Salam Juhmani [1], Chiara Facca [1], Andrea Augusto Sfriso [1], Piero Franzoi [1], Luca Scapin [1], Andrea Bonometto [2], Emanuele Ponis [2], Federico Rampazzo [2], Daniela Berto [2], Claudia Gion [2], Federica Oselladore [2], Federica Cacciatore [2] and Rossella Boscolo Brusà [2]

1   Dipartimento di Scienze Ambientali, Informatica e Statistica (DAIS), Università Ca' Foscari Venezia, Via Torino 155, 30170 Mestre (Ve), Italy; alessandro.buosi@unive.it (A.B.); yari.tomio@unive.it (Y.T.); abdulsalam.juhmani@unive.it (A.-S.J.); facca@unive.it (C.F.); asfriso@unive.it (A.A.S.); pfranzoi@unive.it (P.F.); luca.scapin@unive.it (L.S.)
2   Istituto Superiore per la Protezione e la Ricerca Ambientale (ISPRA), Loc. Brondolo, 30015 Chioggia (Ve), Italy; andrea.bonometto@isprambiente.it (A.B.); emanuele.ponis@isprambiente.it (E.P.); federico.rampazzo@isprambiente.it (F.R.); daniela.berto@isprambiente.it (D.B.); claudia.gion@isprambiente.it (C.G.); federica.oselladore@isprambiente.it (F.O.); federica.cacciatore@isprambiente.it (F.C.); rossella.boscolo@isprambiente.it (R.B.B.)
*   Correspondence: sfrisoad@unive.it; Tel.: +39-041-234-8529

**Abstract:** Since the 1960s, the Venice Lagoon has suffered a sharp aquatic plant constriction due to eutrophication, pollution, and clam fishing. Those anthropogenic impacts began to decline during the 2010s, and since then the ecological status of the lagoon has improved, but in many choked areas no plant recolonization has been recorded due to the lack of seeds. The project funded by the European Union (LIFE12 NAT/IT/000331-SeResto) allowed to recolonize one of these areas, which is situated in the northern lagoon, by widespread transplantation of small sods and individual rhizomes. In-field activities were supported by fishermen, hunters, and sport associations; the interested surface measured approximately 36.6 km$^2$. In the 35 stations of the chosen area, 24,261 rhizomes were transplanted during the first year, accounting for 693 rhizomes per station. About 37% of them took root in 31 stations forming several patches that joined together to form extensive meadows. Plant rooting was successful where the waters were clear and the trophic status low. But, near the outflows of freshwater rich in nutrients and suspended particulate matter, the action failed. Results demonstrate the effectiveness of small, widespread interventions and the importance of engaging the population in the recovery of the environment, which makes the action economically cheap and replicable in other similar environments.

**Keywords:** aquatic angiosperms; transitional waters; environmental restoration; ecological status; Venice Lagoon

## 1. Introduction

Seagrasses, and more generally the aquatic angiosperms [1], play a key role both in marine and transitional water ecosystems (TWS). These structuring plants are considered environmental engineers to which multiple functions are associated [2]. From the morphological point of view, they reduce the impact of winds and tides on sediment resuspension, favor the sedimentation of the suspended particulate, and contrast the erosion of the seabed and the morphological structures of

shallow bottoms [2]. Aquatic angiosperms characterize the bottoms of habitat 1150* (coastal lagoons) and 1140 (muddy or sandy bottoms emerging during low tide) *sensu* Habitat Directive 92/43/EEC, contribute to $CO_2$ sequestration [3], and form the natural habitat for the biological communities [4] providing shelter and food for fish and macrofaunal organisms [5,6]. However, coastal areas and TWS are often very degraded and aquatic angiosperms have disappeared or are in rapid regression; that is mainly due to anthropogenic impacts such as eutrophication or pollution [7]. This was the case of the Venice Lagoon [8,9] and the lagoons of the Po Delta [10,11]. In both the TWS, aquatic angiosperms suffered from two main impacts: the overgrowth of nuisance macroalgae due to the eutrophication increase during the 1960s–1980s and the harvesting of the Manila clam *Ruditapes philippinarum* (Adams and Reeve) with hydraulic or mechanical rakes. Those activities destroyed the bottoms, uprooted the plants, and resuspended considerable quantities of sediments reducing water transparency and the growth of the plants which had survived [12].

Currently, the trophic conditions of the lagoons and ponds of the Po Delta are still bad/poor, because these environments are strongly affected by the waters of the Po River that drains the Po Valley [10,11]. Furthermore, due to the high trophic conditions, bivalves are abundant, clam fishing activities occur on a large scale, and water remains turbid. The aquatic angiosperms recorded in the past have disappeared and have been replaced by tionitrophilic macroalgae. The dominant taxa are Ulvaceae and the non-native Rhodophyceae, *Agarophyton vermiculophyllum* (Ohmi) Gurgel, J.N. Norris *et* Fredericq and *Solieria filiformis* (Kützing) P.W. Gabrielson.

In the Venice Lagoon, the effects of anthropogenic impacts have decreased since 2011 and the ecological status has started to improve [13]. Macroalgal biomass decreased significantly before the period of intense clam fishing [9] and in the last decade the dominant algal species have also changed. Ulvaceae have been largely replaced by several taxa of good-high ecological value, especially Rhodophyceae, and aquatic angiosperms are recolonizing the lagoon bottoms [7,13].

However, in all the basins of the Po Delta and in many choked areas of the Venice Lagoon plant recolonization was hampered by the lack of seeds. To favor the recolonization in the Venice Lagoon, the European Union funded the restoration project (LIFE12NAT/IT/000331—SeResto; www.lifeseresto.eu). The objective was the recolonization of the northern basin by small diffuse triggers of aquatic angiosperms which are typical of that environment: *Cymodocea nodosa* (Ucria) Ascherson, *Zostera marina* Linnaeus, *Zostera noltei* Hornemann, and *Ruppia cirrhosa* (Petagna) Grande.

In order to provide useful information for the project replication in similar TWS, this paper reports the results of the transplanting activities after the first year of plant rooting, the most common environmental parameters and nutrient concentrations in the various environmental matrices (water column, surface sediments, and suspended particulate matter (SPM)), the transplantation methods, and the most suitable environmental conditions to ensure the success of species rooting and spread.

## 2. Materials and Methods

### 2.1. Study Area

The transplants of aquatic angiosperms took place in an area measuring approximately 36.6 km$^2$ (Figure 1) situated in the northern basin of the Venice Lagoon (sexagesimal coordinates: 45°30′–34′ N, 12°27′–33′ E).

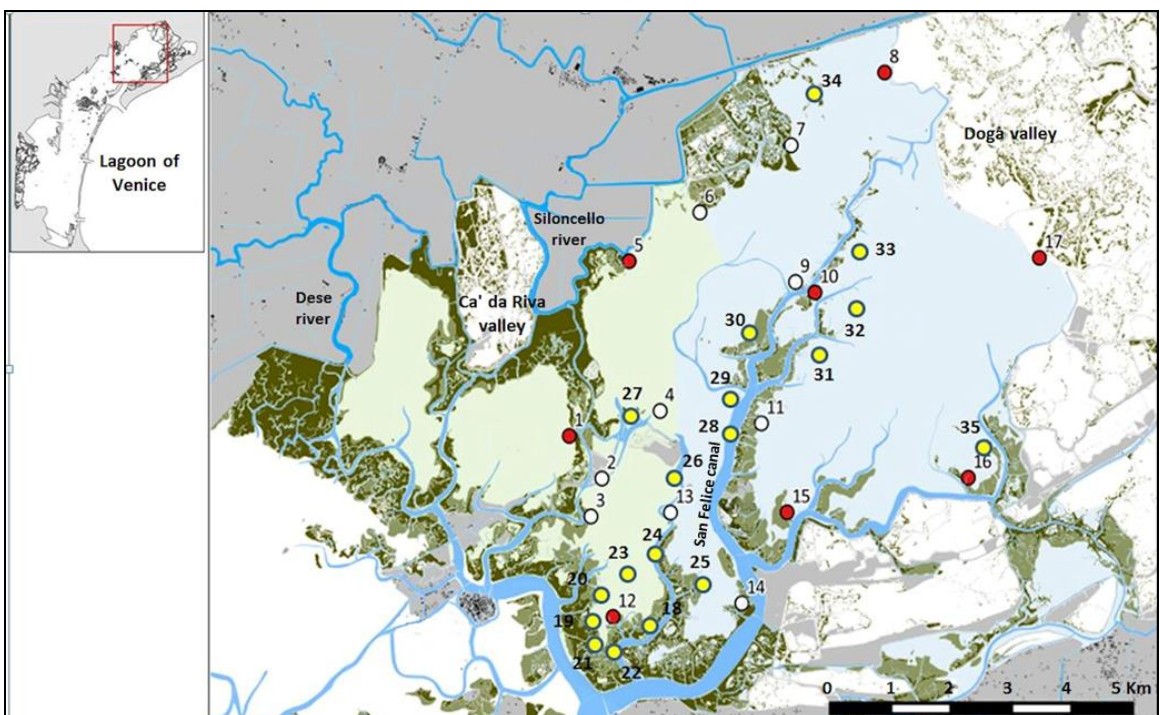

**Figure 1.** Map of the sampling area with the 35 transplanting stations. In red and white, the 17 stations transplanted in spring 2014. The stations in red were monitored monthly for one year for environmental parameters. In yellow, the 18 stations transplanted in spring 2015.

Thirty-five stations characterized by shallow waters were identified in the study area, along the edges of the salt marshes and lagoon canals. The transplanting area is naturally divided by a deep canal (San Felice) which flows in a south-northern direction. On the east side of the lagoon, bottoms are shallower, there is no source of freshwater, and the trophic status is low. On the contrary, the trophic status of the west side is quite high due to the waters of some branches of the Silone river (flow $5 \text{ m}^3 \text{ s}^{-1}$) and the Sile river overflows in rainy periods (flow rates: from a few $\text{m}^3 \text{ s}^{-1}$ to $70 \text{ m}^3 \text{ s}^{-1}$), which have an average frequency of 8–9 events per year [14]). The waters, rich in nutrients and suspended particulate, favor the growth of tionitrophilic algae and trigger phytoplankton blooms that hamper plant rooting.

*2.2. Angiosperm Transplants*

In spring 2014, transplanting occurred in 17 stations of $100 \text{ m}^2$ ($10 \times 10$ m) and in spring 2015, in additional 18 stations (Figure 1). In the initial phase, the transplants took place using sods that were approximately 30 cm in diameter which were collected with a manual corer and arranged in groups of three for a total of nine sods per station, following the scheme in Figure 2. In order to avoid damaging the bottom, all operations were carried out by remaining on board of flat local boats or by divers. The depth of the intervention area was generally less than one meter on the average tide level and the boats were used during high tide to reach even the shallowest areas that emerge at low tide. Angiosperm sods and rhizomes were supplied by managers of closed fishing ponds (Dogà valley and Ca' da Riva valley) where ecological conditions are high and aquatic angiosperms are abundant. Hundreds of full-grown rhizomes were transplanted individually at each station using pliers with a handle of approximately 1 m length.

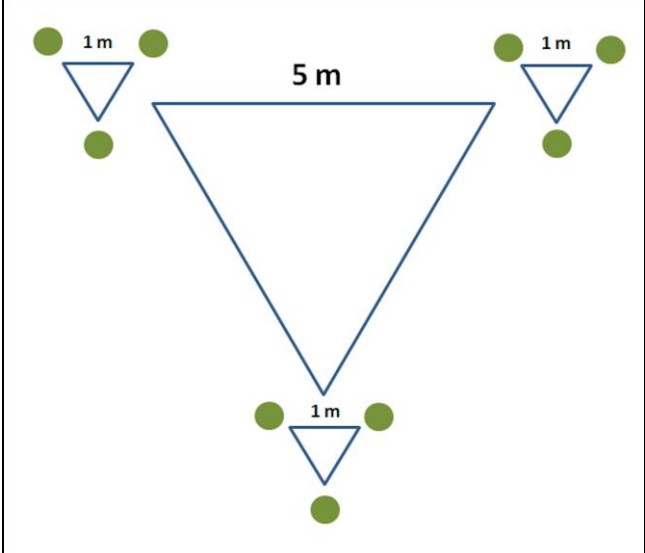

**Figure 2.** Scheme of sod transplants.

### 2.3. Angiosperm Monitoring

Sod rooting was verified on a monthly basis during the first months of spring 2014 and 2015 in order to replace those that had not taken root. The check was repeated, in summer and autumn. In addition, 100 rhizomes of *Zostera marina* that had been planted at each station were carefully monitored during the year to check the success rate of the single rhizome transplant.

Patch measurements reported in this paper refer to monitoring carried out one year after transplants. The growth of sods and rhizomes was mainly circular, therefore during each survey the average growth of patches was recorded by measuring their diameters and took into account only diameters >20 cm.

### 2.4. Physico-Chemical Parameter Determination

Before angiosperm transplants, the environmental conditions (physico-chemical parameters and nutrient concentrations) of the water column and surface sediments were monitored once in all the 35 stations. Out of them, eight stations (1, 5, 8, 10, 12, 15, 16, and 17), representative of the environmental conditions of the whole area, were chosen for the monthly detection of the environmental parameters of water, surface sediments, and the settled particulate material (SPM) collected by sedimentation traps placed on the bottom. The ecological status of the eight stations was also determined by sampling all the macrophytes (twice a year), fish fauna (twice a year), and the benthic macrofauna (once a year) according to the procedures used for the application of the macrophyte quality index (MaQI) [15], habitat fish biotic index (HFBI) [16], and multivariate-AZTI marine biotic index (M-AMBI) [17].

### 2.5. Statistical Analyses

The average values of (a) thirty eight physico-chemical parameters in the water column, surface sediments, and SPM collected monthly for one year in eight stations (1, 5, 8, 10, 12, 15, 16, and 17), (b) seven aquatic angiosperm variables: number, diameter, and growth of surviving sods and rhizomes, and (c) the results of the application of three indices of ecological status determined at the end of the sampling year were analyzed. The Shapiro–Wilk test differentiated non-normal data, and then Spearman's one-way ANOVA coefficients were calculated and summarized in Table 1. Significant values ($p < 0.05$) are marked in red.

**Table 1.** Spearman's non-parametric coefficients between sod/rhizome variables and physico-chemical parameters and nutrient concentrations in the water column and surface sediments. Significant values ($p < 0.05$) are in red. Legend:, Temp = water temperature, Transp = water transparency, Salin = salinity, %DO = percentage of dissolved oxygen, Eh = redox potential, w = water, s = sediment, Light-S = light at a depth of $-5$ cm, Light-B = light at bottom, FPM = filtered particulate matter, Si = silicates, RP = reactive phosphorus, $NH_4^+$ = ammonium, $NO_2^-$ = nitrites, $NO_3^-$ = nitrates, DIN = dissolved inorganic nitrogen, M-biom = macroalgal biomass, M-cover = macroalgal cover, Chl-*a* tot = total Chlorophyll-*a*, TP = total phosphorus, IP = inorganic phosphorus, OP = organic phosphorus, TN = total nitrogen, TC = total carbon, OC = organic carbon, SPM = settled particulate matter, part = particulate.

| Spearman's Non Parametric Coefficients | | | | | | | | | | | |
| --- | --- | --- | --- | --- | --- | --- | --- | --- | --- | --- | --- |
| | Temp | Depth | Transp | Salin | %DO | pHw | Ehw | pHs | Ehs | Light-S | Light-B | FPM |
| Survived sods | 0.03 | −0.21 | −0.09 | 0.63 | −0.33 | 0.43 | 0.18 | 0.30 | 0.39 | −0.34 | 0.48 | −0.31 |
| Sod diameter | −0.29 | −0.27 | −0.17 | 0.73 | −0.54 | 0.44 | 0.00 | 0.41 | 0.46 | −0.56 | 0.54 | −0.29 |
| Sod growth | −0.29 | −0.27 | −0.17 | 0.73 | −0.54 | 0.44 | 0.00 | 0.41 | 0.46 | −0.56 | 0.54 | −0.29 |
| Survived rhizomes | 0.07 | −0.39 | −0.29 | 0.66 | −0.32 | 0.42 | 0.44 | 0.51 | 0.59 | 0.00 | 0.71 | −0.27 |
| Rhizome diameter | −0.14 | −0.06 | 0.11 | 0.75 | −0.71 | 0.69 | 0.02 | 0.34 | 0.20 | −0.62 | 0.24 | −0.67 |
| Rhizome growth | −0.07 | −0.12 | 0.05 | 0.67 | −0.57 | 0.69 | 0.19 | 0.36 | 0.21 | −0.50 | 0.40 | −0.60 |
| Patch surface | 0.00 | −0.32 | −0.22 | 0.63 | −0.34 | 0.39 | 0.29 | 0.37 | 0.51 | −0.22 | 0.63 | −0.20 |
| MaQI | 0.34 | 0.40 | 0.56 | 0.27 | −0.38 | 0.87 | 0.39 | 0.46 | −0.38 | −0.08 | −0.04 | −0.95 |
| M-AMBI | 0.02 | 0.14 | 0.17 | −0.71 | 0.26 | 0.10 | 0.24 | −0.31 | −0.67 | 0.21 | −0.19 | 0.07 |
| HFBI | 0.33 | 0.10 | 0.19 | 0.26 | 0.26 | 0.19 | 0.02 | 0.00 | 0.00 | −0.05 | −0.17 | −0.43 |
| | Si | RP | $NH_4^+$ | $NO_2^-$ | $NO_3^-$ | DIN | M-biom | M-cover | Chl-*a* tot | | | |
| Survived sods | −0.65 | −0.65 | −0.74 | −0.30 | −0.50 | −0.56 | −0.33 | −0.72 | −0.33 | | | |
| Sod diameter | −0.61 | −0.61 | −0.61 | −0.17 | −0.27 | −0.34 | −0.27 | −0.66 | −0.41 | | | |
| Sod growth | −0.61 | −0.61 | −0.61 | −0.17 | −0.27 | −0.34 | −0.27 | −0.66 | −0.41 | | | |
| Survived rhizomes | −0.68 | 0.71 | 0.56 | −0.41 | −0.56 | −0.56 | −0.39 | -0.68 | −0.51 | | | |
| Rhizome diameter | −0.57 | −0.60 | −0.83 | −0.42 | −0.61 | −0.68 | −0.24 | −0.66 | −0.54 | | | |
| Rhizome growth | −0.71 | −0.74 | −0.79 | −0.45 | −0.60 | −0.64 | −0.41 | −0.81 | −0.40 | | | |
| Patch surface | −0.66 | −0.66 | −0.56 | −0.22 | −0.41 | −0.44 | −0.47 | −0.71 | −0.37 | | | |
| MaQI | −0.69 | −0.69 | −0.70 | −0.95 | −0.88 | −0.84 | −0.37 | −0.61 | −0.36 | | | |
| M-AMBI | 0.12 | 0.05 | 0.48 | −0.10 | 0.21 | 0.31 | −0.18 | −0.02 | 0.62 | | | |
| HFBI | −0.17 | −0.19 | −0.76 | −0.48 | −0.67 | −0.76 | 0.26 | −0.19 | −0.24 | | | |
| | Fines | Density | Moisture | Porosity | TP sed | IP sed | OP sed | TN sed | TC sed | OC sed | | |
| Survived sods | −0.20 | 0.39 | −0.39 | −0.39 | −0.73 | −0.37 | −0.86 | −0.65 | 0.78 | −0.05 | | |
| Sod diameter | −0.15 | 0.49 | −0.46 | −0.37 | −0.81 | −0.49 | −0.90 | −0.65 | 0.75 | −0.10 | | |
| Sod growth | −0.15 | 0.49 | −0.46 | −0.37 | −0.81 | −0.49 | −0.90 | −0.65 | 0.75 | −0.10 | | |
| Survived rhizomes | −0.07 | 0.44 | −0.51 | −0.49 | −0.56 | −0.20 | −0.83 | −0.63 | 0.68 | −0.20 | | |
| Rhizome diameter | −0.18 | 0.74 | −0.66 | −0.57 | −0.92 | −0.61 | −0.84 | −0.61 | 0.66 | −0.22 | | |
| Rhizome growth | −0.26 | 0.57 | −0.50 | −0.43 | −0.93 | −0.64 | −0.83 | −0.54 | 0.66 | −0.02 | | |
| Patch surface | −0.24 | 0.39 | −0.41 | −0.32 | −0.66 | −0.29 | −0.85 | −0.65 | 0.75 | −0.10 | | |
| MaQI | −0.35 | 0.48 | −0.40 | −0.47 | −0.64 | −0.63 | −0.25 | −0.01 | −0.07 | 0.12 | | |
| M-AMBI | −0.07 | −0.52 | 0.64 | 0.64 | 0.17 | −0.31 | 0.74 | 0.93 | −0.71 | 0.68 | | |
| HFBI | 0.21 | 0.26 | −0.29 | −0.60 | −0.24 | 0.00 | −0.33 | −0.37 | 0.37 | −0.20 | | |
| | SPM | TP part | IP part | OP part | N part | TC part | OC part | | | | | |
| Survived sods | −0.01 | −0.51 | −0.31 | −0.74 | −0.26 | 0.03 | −0.16 | | | | | |
| Sod diameter | 0.00 | −0.56 | −0.27 | −0.81 | −0.15 | 0.17 | −0.17 | | | | | |
| Sod growth | 0.00 | −0.56 | −0.27 | −0.81 | −0.15 | 0.17 | −0.17 | | | | | |
| Survived rhizomes | 0.05 | −0.68 | −0.37 | −0.78 | −0.32 | −0.15 | −0.34 | | | | | |
| Rhizome diameter | −0.30 | −0.68 | −0.63 | −0.75 | −0.30 | 0.30 | −0.29 | | | | | |
| Rhizome growth | −0.31 | −0.57 | −0.57 | −0.71 | −0.24 | 0.29 | −0.14 | | | | | |
| Patch surface | 0.10 | −0.61 | −0.32 | −0.76 | −0.05 | 0.12 | −0.12 | | | | | |
| MaQI | −0.64 | −0.32 | −0.91 | −0.22 | −0.56 | 0.21 | −0.16 | | | | | |
| M-AMBI | −0.67 | 0.74 | 0.07 | 0.79 | 0.29 | 0.29 | 0.55 | | | | | |
| HFBI | −0.19 | −0.17 | −0.21 | −0.31 | −0.86 | −0.55 | −0.45 | | | | | |

The principal component analysis (PCA) was applied to the same log-transformed data after removing redundancies to visualize the variances and associations between parameters, aquatic angiosperm variables, and index values of the eight stations. Results are reported in a bi-plot. Data were processed by Statistica software, release 10 (StatSoft Inc. Tulsa, OK, USA).

## 3. Results

Figure 3a shows per station, the average diameter of the patches formed by sods and rhizomes 12 months after transplants. The growth was very variable due to the different environmental conditions of the stations. The maximum average diameter of the patches was 143 cm at station 14. Patches over 100 cm in diameter were also found in six other stations (9, 11, 12, 15, 18, 33). In three stations, the sods disappeared completely despite frequent new transplants. In the other stations, the diameter increase was intermediate.

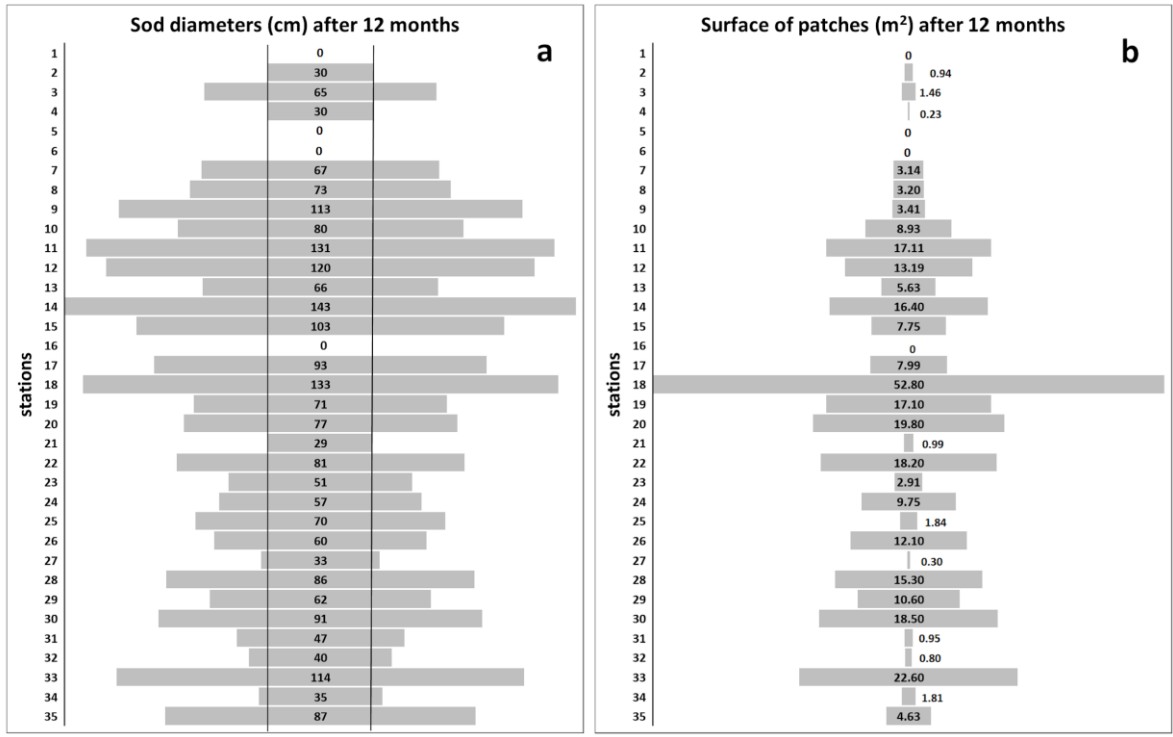

**Figure 3.** (**a**) Average diameters of the patches. Vertical bars represent the diameter of the transplanted sods; (**b**) Total surface of patches, formed by the growth of sods and rhizomes at the 35 sampling sites after 12 months. Only patches with a diameter >20 cm were considered.

The total surface of the patches is shown in Figure 3b. In most stations, there was no correspondence between the diameter and the surface of the patches. The patch surface, in fact, after one year did not distinguish between the sods and rhizomes that grew mixed. In some cases, the total patch surface was the result of a large quantity of patches, in other cases its size depended only on the number of those that had survived. Therefore, results were very different. The maximum growth occurred at station 18 with a colonization of 52% (52 m$^2$ out of 100 m$^2$) and a patch diameter that was among the biggest (133 cm). In contrast, at station 14, just less than 1 km away, the coverage was only 16.4%, even though the diameter of the patches was bigger (143 cm). No angiosperm rooting was observed at stations 1, 5, 6, and 16. Stations 1, 5, and 6 were affected by river outflows, whereas station 16 showed a higher trophic level than the other stations, which was probably due to the proximity of the Cavallino urban center.

The main metrics of sods and rhizomes one year after transplants are shown in Figure 4. The number of survived sods was, on average, 72 ± 38% with high differences between the stations. All the transplanted sods took root in 20 stations out of 35. No angiosperm rooting was found in four stations (1, 5, 6, and 16), whereas in the other stations, their rooting varied widely. On average, rhizome rooting was lower (37 ± 25%) than that of sods, but the number of single rhizome transplants per station was much higher (637) than the number of sods, and so was their potential for developing new rooting.

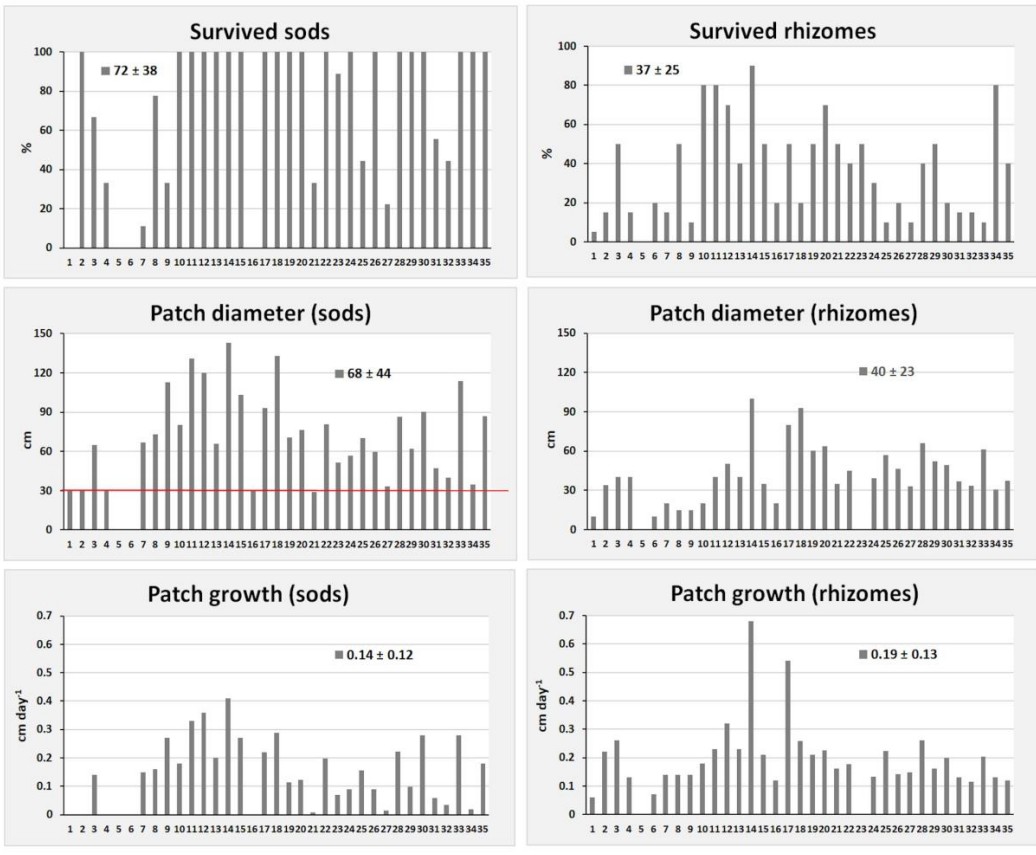

**Figure 4.** Percentage, patch diameter, and daily growth of survived sods and rhizomes.

The average diameter of the patches formed by the surviving sods was 68 ± 44 cm, whereas rhizomes exhibited, on average, a diameter of 40 ± 23 cm, a value very similar to that of sods if we take into account that the diameter of transplanted sods was 30 cm. Moreover, some rhizomes took root also at stations 6 and 16, whereas at station 23 some survived rhizomes formed patches with a diameter shorter than 20 cm. The stations 14 and 18 produced the largest patches.

By comparing growth, rhizomes showed values higher than sods: 0.19 ± 0.13 cm day$^{-1}$ vs. 0.14 ± 0.12 cm day$^{-1}$. It is interesting to observe that the rhizomes transplanted at station 14 showed the highest growth with 0.68 cm day$^{-1}$ and formed patches of approximately 1 m in diameter.

The analysis of nutrient concentrations in the samples of surface sediment collected in all 35 stations during the spring transplants shows that the area is divided into two parts: the one situated northwest of the San Felice canal is characterized by high nutrient concentrations due to river outflows; the other which is on the southeastern side of the canal is characterized by a higher seawater renewal and no river outflow (Figure 5).

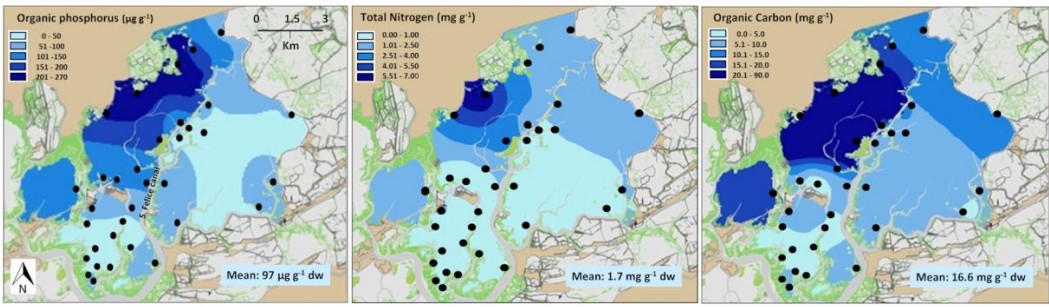

**Figure 5.** Concentration of organic phosphorus, total nitrogen, and organic carbon in surface sediments of the transplant area. Black dots represent transplant stations.

Stations 5 and 6 were the most affected by high nutrient concentrations that favored the growth of tionitrophilic macroalgae and phytoplankton blooms which prevent the rooting of aquatic angiosperms. In the northwestern area of the San Felice canal, the concentration of organic phosphorus (OP), total nitrogen (TN), and organic carbon (OC) were one order of magnitude higher than in the southeastern area.

Similarly, the analysis of the mean concentrations of nutrients in the water column of the eight stations selected for monthly monitoring shows that reactive phosphorus (RP), dissolved inorganic nitrogen (DIN), and silicates (Si) were significantly higher at stations 5 and 1 than in the others (Figure 6).

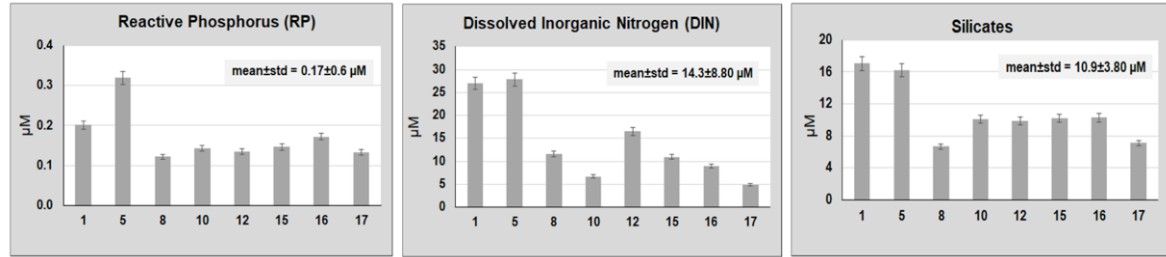

**Figure 6.** Concentration of reactive phosphorus, dissolved inorganic nitrogen, and silicates in the eight stations monitored monthly during one year.

On average, RP showed a concentration of $0.17 \pm 0.6$ μM with the highest value at station 5 (0.32 μM) near the mouth of the Siloncello river. The same results were found for DIN that averagely was $14.3 \pm 8.8$ μM, but at station 5, it was approximately twice as high (27.8 μM). Silicates were slightly higher at station 1 (17.0 μM) and the mean value was $10.9 \pm 3.8$ μM.

Salinity confirms the influence of the Siloncello freshwaters. Stations 5 and 1 exhibited values of approximately 21.0 and 22.4 psu, respectively, against 26.8 to 29.2 recorded in the others (Figure 7). Total chlorophyll-*a* (Chl-*a*) also showed the highest value (approximately 1.3 μg L$^{-1}$), although, on average, it was very low ($0.94 \pm 0.29$ μg L$^{-1}$). The particulate collected by sedimentation traps (SPM) highlighted the environmental differences between the stations more clearly than Chl-*a*, because the sedimentation rates were collected and recorded constantly during the whole sampling year. Also, this parameter showed the highest value at station 5 (635 g dw m$^{-2}$ d$^{-1}$) accounting for approximately 232 kg dw m$^{-2}$ y$^{-1}$. On the contrary, extremely low values were found at stations 8 and 17, where aquatic angiosperms had widely taken root.

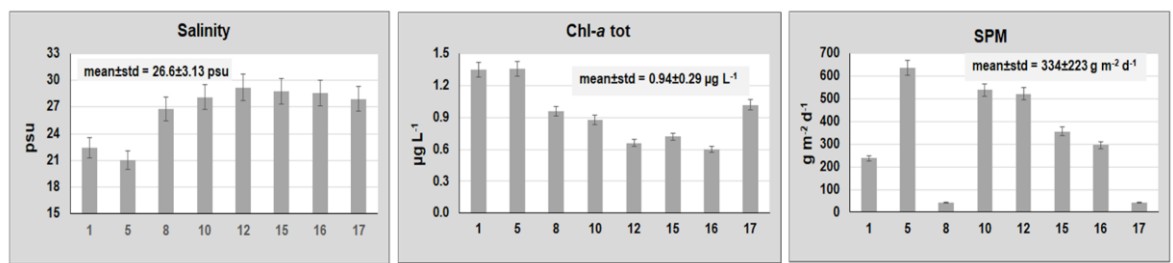

**Figure 7.** Values of salinity, total chlorophyll-*a* (Chl-*a* tot), and settled particulate matter (SPM) collected by sedimentation traps).

Two other important parameters that marked the differences between the stations were the percentage of sediment fraction <63 μm (fines) and the amount of dry sediment per volume unit (dry density) of surface sediments. Grain-size was >70% in all the stations (Figure 8), but with a different degree of compactness. Dry density values were overall very low ($0.43 \pm 0.14$ g dw cm$^{-3}$), but the lowest were recorded at station 5 (0.16 g dw cm$^{-3}$) and station 1 (0.32 g dw cm$^{-3}$), where aquatic angiosperms had not taken root.

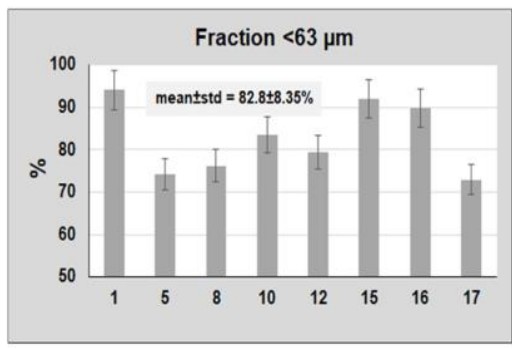 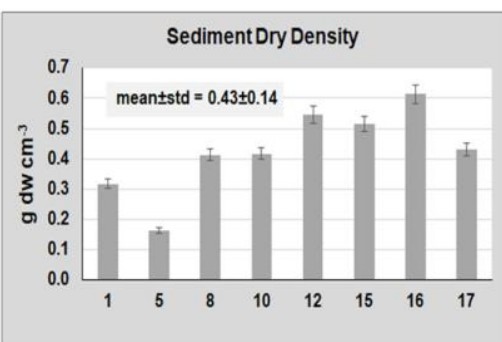

**Figure 8.** Percentage of fines (fraction < 63 μm) and dry density of surface sediments.

The non-parametric Spearman's coefficients calculated using 38 physico-chemical parameters, seven macrophyte variables, and three indices of ecological status in the eight stations showed that freshwaters coming from the river Siloncello affected the salinity of the water column seriously, deteriorating the ecological conditions and hindering the angiosperm rooting (Table 1). In fact, sod diameter (cm), sod surface (m$^2$), and sod growth (cm day$^{-1}$) were positively correlated to salinity, whereas rhizome survival (% yr$^{-1}$) was positively correlated to light availability at the bottom (Light-B), and inversely to SPM and the concentration of organic phosphorus (OP) in surface sediments.

Moreover, rhizome growth was counteracted by the concentrations of Si, RP, and ammonium (NH$_4^+$) in the water column by TP and OP in surface sediments and SPM. Macroalgal cover (M-cover) was also inversely and significantly correlated to angiosperm growth. The accumulation of tionitrophilic macroalgae, especially Ulvaceae and Gracilariaceae, on sod and rhizome transplants prevented their growth. These macroalgae have a much higher growth rate [18] than angiosperms [19]. The same inverse correlations were also found between sod growth and TP, OP in surface sediments, and OP in SPM. The indices of ecological status: MaQI, M-AMBI, and HFBI showed that the correlations with the physico-chemical parameters in the water column had a similar trend, whereas in surface sediments the correlations were not always clear, especially when M-AMBI was applied.

The principal component analysis (PCA) applied to the whole set of parameters and variables, after the exclusion of redundancies, confirmed the above relationships (Figure 9). About 66% of the total variance was explained by the first two components that clearly separate the variables of the aquatic angiosperms and most of the environmental parameters into two major groups. In the two minor groups, the variables of the first two components were intermediate. In group 1, both sod and rhizome rooting were associated with salinity, sediment density, sediment total carbon, sediment and water pH, and light availability at the bottom. The ecological quality ratio (EQR) of the indices of ecological status of the biological elements, macrophytes (MaQI) and fish fauna (HFBI), were also associated to aquatic angiosperm variables and the same environmental parameters.

In group 2, most of the nutrient concentrations in the water column, surface sediments, and SPM were plotted together with sediment moisture and porosity, water temperature, the %DO, Chl-*a*, FPM, and M-AMBI. Groups 3 and 4 highlighted some parameters which were intermediate between groups 1 and 2.

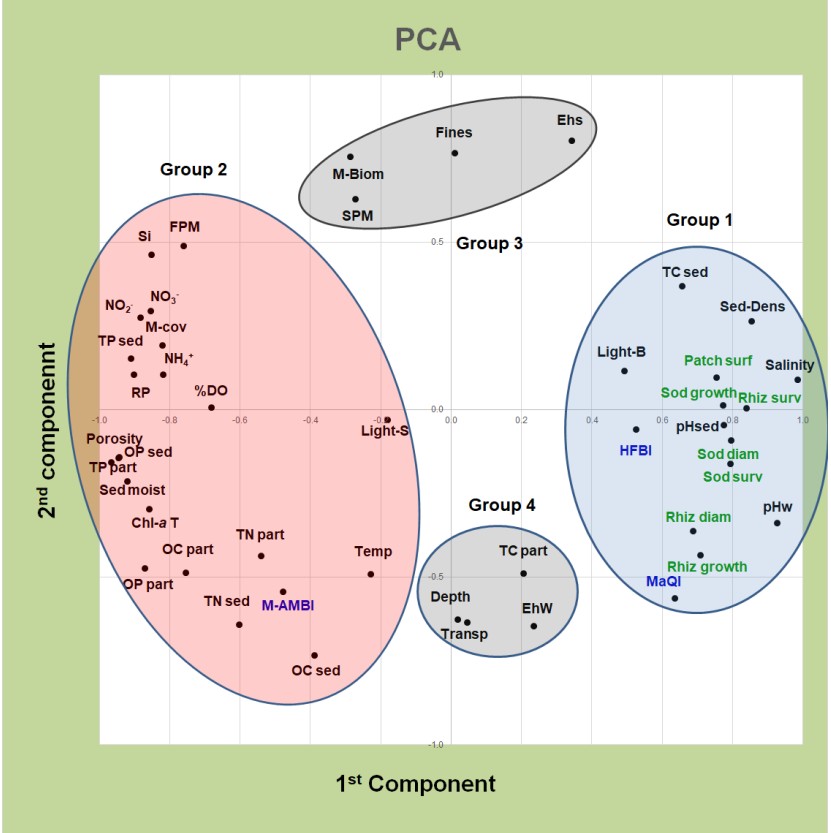

**Figure 9.** Principal component analysis (PCA) between physico-chemical parameters and macrophyte variables. Legend: Rhiz = rhizomes, Surv = survived, Diam = diameter, surf = surface, Temp = water temperature, Transp = water transparency, %DO = percentage of dissolved oxygen, w = water, sed = sediment, dens = density; moist = moisture, Light-S = light at a depth of −5 cm, Light-B = light at bottom, FPM = filtered particulate matter, Si = silicates, RP = reactive phosphorus, $NH_4^+$ = ammonium, $NO_2^-$ = nitrites, $NO_3^-$, = nitrates, DIN = dissolved inorganic nitrogen, M-Biom = macroalgal biomass, M-cov = macroalgal cover, Chl-*a* T = total chlorophyll-*a*, TP = total phosphorus, OP = organic phosphorus, TN = total nitrogen, TC = total carbon, OC = organic carbon, SPM = settled particulate matter, part = particulate.

## 4. Discussion

The results of this study highlight the potential of small primers, especially diffused transplants of single rhizomes, for a rapid colonization of areas that have the suitable ecological conditions. The transplants of aquatic angiosperms by single rhizomes do not require large efforts and expertise and favors the diffusion of plants throughout the area. Therefore, the involvement of motivated groups of people such as fishermen, hunters, and sport associations who frequently spend part of their time in the lagoon was a great achievement for lagoon recovery. In the past, several transplants of sods measuring some square meters [20–22] were carried out at the edges of canals or in relatively deep areas. Such operations required great efforts, and also the use of large boats and mechanical means that cannot operate in shallow waters. The Project Life12 NAT/IT/000331—SERESTO (www.lifeseresto.eu) tested the possibility of recolonizing a large and shallow lagoon area (approximately 36.6 km$^2$), where the aquatic angiosperms had almost completely disappeared, through small, widespread transplants. The proposed strategy achieved a large-scale restoration through the transplants of a small number of plants (approximately 40 m$^2$ during the entire project), with advantages in terms of costs and impact on the donor sites.

Intense monitoring of plant growth and environmental parameters over the whole area during one year allowed to understand the reasons behind achievements and/or failures. Results showed that aquatic angiosperms took roots only in areas where nutrient concentrations were not too high and where the tionitrophilic macroalgae, especially Ulvaceae, Cladophoraceae, and Gracilariaceae prevailed, both sods and rhizomes chocked and died. Angiosperm transplants failed near river outflows because waters were turbid and rich in nutrients. The choice to transplant angiosperms as well in these areas and to monitor the environmental parameters allowed to identify a series of critical values that must be taken into account should these techniques be successfully applied to other areas. First of all, as reported in other studies [7], the presence and growth of aquatic angiosperms depended on the species considered, degree of trophic conditions, and clearness of the waters. The lagoon areas we studied are colonized by five species: three seagrasses (*Cymodocea nodosa* Ucria (Ascherson), *Zostera marina* Linnaeus, and *Zostera noltei* Hornemann) and two aquatic angiosperms (*Ruppia cirrhosa* Petagna (Grande) and *Ruppia maritima* Linnaeus) [23]. All of them have different ecological features. *Cymodocea nodosa* is a subtropical species that colonizes areas characterized by high salinity, coarse grain sediments, and low nutrient concentrations. It prefers the areas which are strongly influenced by marine waters near the lagoon mouths or high-flow canals, but it can also be present in choked areas [7,23] provided there are the conditions quoted above. However, in the shallow choked areas, where water renewal is low, it reaches much smaller sizes than in areas vivified by the sea. *Zostera marina* prefers areas characterized by fine sediments and high water exchange, because it is a species very sensitive to high temperatures. It reaches the highest growth in April–May, afterwards it decreases and disappears almost completely when the water temperature remains above 26–28 °C for long periods. *Zostera noltei* and *Ruppia cirrhosa* prefer shallow choked areas. The former grows mainly along saltmarsh edges and the latter in any shallow bottom, provided the waters are clear. Both species colonize areas with fine sediments where temperature and salinity are very variable, but *Ruppia cirrhosa* prefers the most choked areas and sediments with a higher percentage of fines. Finally, *Ruppia maritima* is the species which most easily adapts to extreme variations in salinity and temperature. It usually grows in the pools of salt marshes, which are in-flowed by new water only during particularly high tides [23].

Our transplants involved the first four species, particularly *Z. marina* and *Z. noltei*, whereas *R. cirrhosa* and *C. nodosa* were only transplanted in some stations with suitable ecological conditions. All the species took root, but *Z. noltei* was the most successful. It rapidly colonized all the saltmarsh edges of the whole transplanting area, where new plants were also produced through natural seed dispersion. *Zostera marina* took root in areas with a negligible tionitrophilic algal biomass, where good water renewal kept temperatures moderate, preventing the formation of sulfides that are very toxic and lead to the species local extinction [24]. Transplants of *Z. marina* carried out in the autumn were a great achievement, because they formed numerous patches, even though in the presence of the high summer temperatures (>26–28 °C), which were also recorded by [25] in the Denmark coasts, they started to regress and in some areas, disappeared altogether. Temperature was the most critical factor for the survival of this species. Some authors [26] who had carried out in-culture experiments fond that the best temperature for the growth of *Z. marina* was <25 °C. At 27 °C, the growth decreased significantly and at higher temperatures it died. This explains why it is widely distributed in the cold waters of the northern part of the Atlantic Ocean and the Pacific Ocean, up to the Arctic Circle. In the Mediterranean Sea, it is a relict species of the ancient Tetide Ocean that colonizes mainly areas characterized by moderate temperatures and salinities [10–25,27] such as the northern Adriatic Sea, the Aegean Sea, the Black Sea, and some areas of the Spanish and French coasts [28]. *Ruppia cirrhosa* was transplanted into 12 stations. In the three decades previous to transplant activities, there was no sign of the presence of this species in the lagoon which is open to tidal expansion, whereas it was abundant in the closed fishing ponds [28,29]. The year after transplanting its spread was explosive, as it rapidly colonized all the choked shallow areas, while no rooting was observed in areas vivified by large canals.

*Cymodocea nodosa* was transplanted only in a station and as expected, its spread was very limited (stations 7 and 17) because of the ecological conditions of the transplant areas.

The establishment and growth of these aquatic angiosperms are strongly affected by the excess of nutrients in the water column, surface sediment, and SPM, especially by the phosphorus compounds (Table 1) as reported by [30]. High sediment moisture and porosity also counteract plant rooting because sediments are less compact, anoxic, and rich of organic nutrients. In addition, sediments characterized by high moisture usually contain a high amount of ammonium [31] and sulfides [32,33] that act as phytotoxins and hamper plant rooting.

Despite the short period considered, the responses of some indices used to assess the ecological status were positive. The macrophyte quality index (MaQI) and habitat fish biotic index (HFBI) only one year after transplanting showed the same positive correlations with the presence of aquatic plants as by Spearman's non-parametric coefficients and PCA. On the contrary, M-AMBI did not respond positively to the presence of aquatic plants. In fact, the benthic macrofauna used for the index application lives inside the sediment, which requires longer time to change its physico-chemical characteristics than the water column. Therefore, we expect that benthic macrofauna will respond to the changes with a delay of a few years.

According to [34–36], pristine TWS should be colonized by aquatic angiosperms. These plants favor the maintenance of good ecological conditions. They control the change of water and sediments pH, favoring the development of calcareous algae, trap large amounts of $CO_2$ [37,38], and prevent sediment erosion and dispersion, favoring sedimentation processes. In addition, angiosperm meadows are natural nurseries where the fish macrofauna finds food and shelter [39–41]. Finally, the small aquatic angiosperms like *Z. noltei* and *Ruppia* spp. that colonize the emerging seabed at low tide are also the best environment for birds such as ducks, flamingos, and herons that feed on both plants and the organisms typical of those areas. Therefore, the recovery of TWS through recolonization of environments with aquatic angiosperms is of fundamental importance. But, the success of transplants largely lies also on the cooperation of the population interested in safeguarding the environments they attend for work or leisure.

**Author Contributions:** Conceptualization, A.S., P.F., A.B. (Andrea Bonometto) and R.B.B.; Formal analysis, A.B. (Alessandro Buosi), Y.T., A.-S.J., A.A.S., L.S., E.P., F.R., D.B., C.G., F.O. and F.C.; Funding acquisition, A.S., A.B. (Andrea Bonometto) and R.B.B.; Investigation, A.S. and P.F.; Methodology, A.S. and A.B. (Andrea Bonometto); Project administration, A.S. and C.F.; Resources, A.S. and A.B. (Andrea Bonometto); Supervision, A.S.; Visualization, A.S. and A.B. (Andrea Bonometto); Writing—original draft, A.S.; Writing—review and editing, A.S., A.B. (Alessandro Buosi), Y.T., A.-S.J., C.F., A.A.S., P.F., L.S., A.B. (Andrea Bonometto), E.P., F.R., D.B., C.G., F.O., F.C. and R.B.B.

**Funding:** This research was funded by the European Union's LIFE+ financial instrument (grant LIFE12 NAT/IT/000331—LIFE SERESTO, which contributes to the environmental recovery of a Natura 2000 site, SIC IT3250031—Northern Venice Lagoon).

**Acknowledgments:** The authors thank Orietta Zucchetta for reviewing the English language and are grateful to the anonymous reviewers that have revised the paper for useful suggestions to improve the manuscript.

**Conflicts of Interest:** The authors declare no conflict of interest.

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
