# Peer review of "Aquatic Angiosperm Transplantation: A Tool for Environmental Management and Restoring in Transitional Water Systems"

_water, doi:10.3390/w11102135_

Round 1

Reviewer 1 Report

Aquatic angiosperm transplantation: a tool for environmental management and restoring in transitional water systems. Adriano Sfriso et al. in “Water”

This is a very basic study/survey conducted in the lagoons of Venice, Italy about the results of transplantations of rizomes and sods of aquatic marcrophytes in an effort to restore water quality and the natural flora and fauna. The transplantations were carried out in 2014 and 2015 and the growth was monitored along with water parameters including nutrient load for a period of an year (I assume). All in all, I find the monitoring was carried out scientifically but not qualify for a research paper in “Water”. This as it is appears as a project report as there is no hypothesis tested or experiments planned other than just monitoring the growth of these transplanted macrophytes. I feel this paper will not be of much interest to the readers and therefore I don’t recommend publication in “Waters”.

A small note to the authors, the manuscript needs English editing. This will be of interest to a local publication.

Author Response

The Project proposal was submitted to European Union after years of in field experiments on the growth and primary production measurements of the transplanted species under different ecological conditions. Results were reported in several papers (1-9). Still the positive results of a pilot experimentation of transplants carried out during two years in other areas of the lagoon provided us with the scientific information needed to proceed with a larger project. All this information was detailed in transplant and monitoring protocols produced during the implementation of the project and reported in a “Practical guide for the recognition of phanerogams and for transplantation actions”(10).

In our opinion disseminating information on best practices for the recovery or degraded transitional water systems through recolonization of those environments with aquatic angiosperms, is of high interest for readers and the replicability in other similar environments. It is also important to note that these recovery activities have a greater chance of success when they involve the population interested in safeguarding the environments they attend for work or leisure.

Finally the  English editing was extensively revised by a person with language familiarity.

Below some papers of my research team on seagrass growth:

1 - Sfriso, A., Marcomini,A. (1997). Macrophyte production in a shallow coastal lagoon. Part I. Coupling with physico-chemical parameters and nutrient concentrations in waters.    Marine Environmental Research, 44: 351-375. 

2 - Sfriso, A., Ghetti, P.F. (1998).  Seasonal variation in the  biomass, morphometric parameters and production of rhizophytes in the lagoon of Venice.  Aquatic Botany, 61: 207-223.    

3 - Sfriso, A., Marcomini,A. (1999). Macrophyte production in a shallow coastal lagoon. Part II. Coupling with sediment, SPM and tissue nutrient concentrations.   Marine Environmental Research, 47: 285-309.

4 -Rigollet,V., Laugier, T., De Casabianca, M.L., Sfriso, A., Marcomini, A. (1999). Seasonal biomass and nutrient dynamic of Zostera marina L. biomass in two mediterranean lagoons: Thau (France) and Venice (Italy).

5 - Zharova, N., Sfriso, A., Voinov, A., Pavoni,B.  (2001).   A  simulation model for the annual fluctuation of the Eelgrass (Zostera marina) biomass in the Venice lagoon.  Aquatic Botany. 70: 135-150.

6 - Sfriso,A., Facca,C., Ceoldo, S. (2004).Growth and production of Cymodocea nodosa (Ucria) Ascherson in the Venice lagoon. In: P. Campostrini (Ed.). Scientific Research and Safeguarding of Venice. CoRiLa. Research Programme 2001-2003. 2002 Results. Multigraf, Spinea, Vol II. pp.  229-236.

7 - Sfriso, A., Facca, C., Ceoldo, S. (2008). Growth and net production of the seagrass Nanozostera noltii (Hornemann) Tomlinson et Posluzny in Venice lagoon. In: P. Campostrini (Ed.). Scientific Research and Safeguarding of Venice. Corila     Research Program 2004-2006,  2006  Results.   IVSLA. Multigraf, Spinea, Vol. VI, pp. 281-291

8 - Zharova N., Sfriso, A., Pavoni, B., Voinov A. (2008).  Analysis of annual fluctuations of C. nodosa in the Venice lagoon: Modelling approach. Ecological Modelling216: 134-144.

9 - Sfriso, A. , Facca, C., Bonometto, A., Boscolo, R. (2014). Compliance of the Macrophyte Quality index (MaQI) with the WFD (2000/60/EC) and ecological status assessment in transitional areas: The Venice lagoon as study case. Ecological Indicators, 46: 536-547.

10 - Sfriso, A., Boscolo, R., Facca, C., Buosi, A., Bonometto, A., Parravicini, M. (2014)  Life Seresto. Habitat 1150* (Coastal Lagoon) recovery by seagrass restoration. A new  strategic approach to meet HD & WFD objectives. Practical guide for the recognition of phanerogams and for transplantation actions. University Ca’ Foscari Venice, DAIS, pp. 1-75.

Reviewer 2 Report

In the manuscript the authors demonstrate that the success of transplantation method and show that the most suitable environmental conditions promote the success of the spread of the various species. It is a well-written and professionally supported manuscript highlights the above mentioned processes in order to provide useful information for the application of these environmental recovery procedures in similar transitional water systems.  The topic may be of great interest for other scientist!

Author Response

I thank the comment of this referee who understood the purpose of the work.

Reviewer 3 Report

I have reviewed the manuscript entitled: ' Aquatic angiosperm transplantation: a tool for environmental management and restoring in transitional water systems.” The objective of the study has been met, and research findings have been thoroughly discussed. The paper is well organized, and the Results and Discussion sections are clearly and concisely written. I think that the article is very interesting and fully deserves to be published.

SPECIFIC COMMENTS:

Introduction

Line: 36-37 – This should have a reference

Line: 37-40 – This should have a reference

Line: 43-44 – This should have a reference

Line: 52-58 – This should have a reference

Line: 65-66 – This should have a reference

Materials and Methods

The methodology of the study is unclear, specifically no information about statistical methods

Line: 94 - (http://www.arpa.veneto.it/dati-ambientali/open-data/file-e-allegati/2017) move to References

Line: 99-100 – „Seventeen stations of 100 m2 (10 x 10 m) were transplanted in the spring 2014 and another 18 stations 99 in spring 2015”. - Fig 1?

Results

Table 2.  - Suggest to move to the supplements.

Discussion

I have no especial comments on the discussion except:

Lines 257-272.  Consider deleting this part; mentioned already in the Introduction and is not fitting in this part of Discussion.

Author Response

I have reviewed the manuscript entitled: ' Aquatic angiosperm transplantation: a tool for environmental management and restoring in transitional water systems.” The objective of the study has been met, and research findings have been thoroughly discussed. The paper is well organized, and the Results and Discussion sections are clearly and concisely written. I think that the article is very interesting and fully deserves to be published.

Introduction

Line: 36-37 – This should have a reference

Line: 37-40 – This should have a reference

Line: 43-44 – This should have a reference

Line: 52-58 – This should have a reference

Line: 65-66 – This should have a reference

The required references were inserted

Materials and Methods

The methodology of the study is unclear, specifically no information about statistical methods

Information on the Statistical methods were inserted: Point 2.5 (Statistical Analyses) on “Materials and Methods” section.

Line: 94 - (http://www.arpa.veneto.it/dati-ambientali/open-data/file-e-allegati/2017) move to References

The web site was moved on the References

Line: 99-100 – „Seventeen stations of 100 m2 (10 x 10 m) were transplanted in the spring 2014 and another 18 stations 99 in spring 2015”. - Fig 1?

Fig. 1 was added

Results

Table 1.  - Suggest to move to the supplements.

We think that this table is important to show the relationship between environmental parameters and macrophyte variables and we prefer to leave it in the text.

Discussion

I have no especial comments on the discussion except:

Lines 257-272.  Consider deleting this part; mentioned already in the Introduction and is not fitting in this part of Discussion.

This part was rewritten and reduced but in our opinion it is important to underline that the success of transplants was largely due to the use of small transplants  with little efforts and low cost as an alternative to those made in the past that required large boats, mechanical means and high costs. In addition, the cooperation of the population interested in safeguarding the environments they attend for work or leisure was the best choice to obtain rapid and positive results in the lagoon recovery. 

Round 2

Reviewer 1 Report

Thanks for the response to my earlier comments. Yes now I see the writing style of the manuscript has improved leaps. But, unfortunately I still believe this is way too basic information... but as the authors claim, if it is going to benefit readers (I am not in complete agreement here) I dont mind recommending publication. But, what is the need for so many figures? Can it be reduced?